# Current and Future Developments in Radiation Oncology Approach for Rhabdomyosarcoma

**DOI:** 10.3390/cancers17101618

**Published:** 2025-05-10

**Authors:** Raquel Dávila Fajardo, Henriette Magelssen, Alison L. Cameron, Tom Boterberg, Henry C. Mandeville

**Affiliations:** 1Department of Radiation Oncology, University Medical Center Utrecht, 3584 CX Utrecht, The Netherlands; 2Princess Máxima Center for Pediatric Oncology, 3584 CS Utrecht, The Netherlands; 3Department of Oncology, Oslo University Hospital, 0450 Oslo, Norway; henma@ous-hf.no; 4Bristol Cancer Institute, University Hospitals Bristol and Weston NHS Foundation Trust, Bristol BS1 3NU, UK; 5Particle Therapy Interuniversity Centre Leuven—PARTICLE, 3000 Leuven, Belgium; tom.boterberg@ugent.be; 6The Royal Marsden Hospital and Institute of Cancer Research, Sutton SM2 5PT, UK; henry.mandeville@rmh.nhs.uk; 7Proton Beam Therapy, University College London Hospitals NHS Foundation Trust, London NW1 2PG, UK

**Keywords:** rhabdomyosarcoma, radiotherapy, advanced radiotherapy techniques, local treatment

## Abstract

Over the last 20 years, the outcomes for rhabdomyosarcoma, the most common pediatric soft tissue sarcoma, have improved, with radiotherapy being an essential part of multimodality therapy for the majority of patients. However, about one-third of rhabdomyosarcoma (RMS) patients will develop loco-regional relapse, and patients with metastatic or relapsed disease have a poor prognosis, so improved local therapies are needed, in addition to better systemic therapies. This review aims to highlight the current and future developments in radiotherapy as part of the management of RMS with the objective of improving outcomes whilst minimizing treatment-related morbidity.

## 1. Introduction

Rhabdomyosarcoma (RMS) is an infrequent soft tissue sarcoma that mostly presents in children (around 60% of cases), while the adult cases carry a poorer prognosis [1,2]. RMS is the commonest of the pediatric soft tissue sarcomas and affects around 40 children (0–14 years) and 50 older teenagers/adults per year in the UK [1,3]. RMS derives from the embryonal mesenchyme and can arise at almost any site within the human body. The most frequent histological subtypes are alveolar (aRMS), embryonal (eRMS), and spindle cell/sclerosing (ssRMS) [4]. Neoadjuvant chemotherapy is used in the majority of patients leading to a response rate (RR) of around 80–85% [5,6,7]. However, despite this chemo-sensitivity, local therapies including radiotherapy, surgery, or a combination of both are needed for optimal long-term local control and the chance of a cure. In Intergroup RMS Study (IRS)-IV, 695/883 (78.7%) patients with intermediate-risk RMS required radiotherapy as part of their primary treatment [8]. Of those with metastatic disease, approximately 75% can achieve remission, yet the vast majority relapse, often at distant sites, with a 3-year event-free survival (EFS) of only 27% reported [9,10]. Unfortunately, at the time of relapse, RMS is often refractory to treatment, with very poor outcomes for those relapsing after previous radiotherapy, as well as for those with metastatic disease, achieving a 5-year overall survival (OS) of less than 20% [11].

In adults, 80% have histological diagnoses comparable with RMS in children, with the predominance of alveolar histology; the remaining 20% of patients (8% of total RMS cases) have a pleomorphic histology [2]. Prior to 2020, no clinical trials in adult RMS had been performed, although a retrospective single-center experience reported that treatment according to pediatric regimens may improve outcomes [1].

Currently, treatment for newly diagnosed patients in the pediatric population is stratified according to age, tumor size, histology (favorable or unfavorable), Intergroup Rhabdomyosarcoma Study (IRS) post-surgical stage, and lymph node involvement. This treatment stratification strategy was adopted in the previous European pediatric Soft tissue sarcoma Study Group (EpSSG) RMS-2005 trial for non-metastatic RMS in children and has proven to be effective in discriminating survival by subgroup [12]. Building on the results from recent studies, the current EpSSG overarching study for children and adults with Frontline and Relapsed RhabdoMyoSarcoma trial (FaR-RMS) incorporates, and is prospectively evaluating, the use of fusion gene status versus histopathological subtyping in the stratification criteria [12,13,14,15,16].

Internationally, three key collaborative groups are conducting clinical trials in pediatric RMS: the EpSSG and the Cooperative Weichteil Sarcoma (CWS) group in Europe and the Children’s Oncology Group (COG) in North America. The EpSSG, for the first time, has incorporated a number of radiotherapy randomizations into the ongoing FaR-RMS trial; the role of preoperative radiotherapy and of dose escalation in patients with unfavorable features that make them more prone to develop a local recurrence are being evaluated [16]. In addition, the effect of the use of radiotherapy in patients with unfavorable metastatic disease is also being investigated in FaR-RMS [16].

An overview of the current ongoing FaR-RMS study with a focus on the radiotherapy aspects related to the treatment of rhabdomyosarcoma is presented.

## 2. Radiotherapy Management

Despite the significant improvements in outcomes for patients in the last 20 years, local control remains the principal challenge in RMS. The local failure analysis from EpSSG RMS 2005 is being undertaken to understand the impact of surgery, radiotherapy, and other treatment and patient factors, following recent COG publications [17]. Radiotherapy is a key component of local therapy for RMS and analyses from the SIOP (International Society of Paediatric Oncology) MMT (Malignant Mesenchymal Tumor) 84, 89, and 95 trials in pediatric RMS supported the more systematic use of radiotherapy that was adopted in the EpSSG RMS 2005 trial [18]. In EpSSG RMS 2005, 86% of patients with localized HR-RMS received radiotherapy, with the trial reporting an increase in 3-year EFS from 55% to 67% for HR patients and from 39% to 56% for node-positive alveolar patients [19,20]. However, local failure was still observed in the majority of relapse cases. It is proposed that the effectiveness of radiotherapy with regard to local control could be improved by modifying the dose and/or the timing of radiotherapy; within FaR-RMS both strategies were investigated, with prospective radiotherapy quality assurance by the SIOP Europe-EORTC QUARTET platform mandated for randomized cases [21,22,23].

### 2.1. Radiotherapy Dose Escalation

Over the previous four decades, there has been an evolution in the radiotherapy approach, with refinements informed by outcomes from different European and U.S. collaborative group studies. In general, treatment doses between 36–55 Gy (conventional fractionation) and 59.4 Gy (hyperfractionated radiotherapy: HFRT) have been used [18,24,25]. However, for RMS, it remains uncertain as to whether a further radiotherapy dose escalation will produce better local control, and this is currently under investigation in FaR-RMS. An overview of the multiarm–multiphase FaR-RMS trial, including a detailed description of the rationale for radiotherapy dose escalation, has been recently published by Chisholm et al. [26].

Previously, adults (>21 y) have been excluded from the majority of collaborative group RMS studies. RMS is less commonly observed in adult patients, but, when it does occur, it is often associated with larger and more unfavorable tumors, which have worse outcomes, including local failure [27]. The FaR-RMS trial is looking to address this through the inclusion of adult patients; RMS patients with an HLFR, defined as those with an unfavorable primary site or those aged 18 years or older, are the cohort being studied in the randomizations of radiotherapy dose escalation, exploring whether these approaches can achieve improved local control (Figure 1).

### 2.2. Timing of Radiotherapy

The conventional approach for the delivery of adjuvant radiotherapy for RMS is postoperatively, after surgical resection. Nevertheless, the use of preoperative radiotherapy has potential advantages over postoperative radiotherapy. Firstly, the radiotherapy target volume definition is more straightforward, and potentially more accurate, with an intact tumor. Secondly, the residual tumor can act as a form of ‘natural spacer’ and reduce the volume of uninvolved normal tissue exposed to higher radiation doses and could potentially reduce the risk of developing secondary malignancies. From a radiobiological perspective, hypoxia is recognized to increase tumor radio-resistance; therefore, as the preoperative state has less hypoxia in the surrounding tissues than that after surgery, there is a clear rationale to support the investigation of preoperative radiotherapy [28]. Experience using preoperative radiotherapy for soft tissue sarcoma has increased over the years in the standard clinical setting [29]. Yet, despite preoperative radiotherapy having being investigated in a number of non-rhabdomyosarcoma STS studies [30,31], the published experience on its use in the context of rhabdomyosarcoma is scarce, with only a small cohort of 17 patients treated with preoperative radiotherapy in the CWS96 study for a bladder/prostate RMS; these patients had a 5-year EFS of 87.7%, consistent with favorable outcomes [32].

Despite preoperative radiotherapy being a possible treatment option in the EpSSG RMS 2005 trial, only a very small number of patients received this, and its effects have not been systematically evaluated. The FaR-RMS trial is investigating the impact of the timing of adjuvant radiotherapy in a randomization comparing local control and the associated toxicities of postoperative and preoperative radiotherapy. A detailed description of the different randomizations in that study was recently published [26].

### 2.3. Radiotherapy to Metastatic Sites

Despite conflicting data as to the influence of radiotherapy to metastatic sites on survival outcomes, the standard-of-care approach for metastatic RMS in international collaborative group studies continues to be a recommendation for the irradiation of all metastatic sites that can be feasibly treated (MTS-2008 registry study for metastatic RMS within RMS 2005 [33]). This differs significantly from the recommendations for adult soft tissue sarcomas, where radiotherapy to metastatic sites is not commonly used in standard clinical practice. In the COG RMS study protocols, patients with more than five metastatic lesions have been categorized as having extensive metastatic disease with radiotherapy applied at week 20. Due to the derived technical challenges, the COG has advised restricting to a maximum of five treatment fields and prioritizing weight-bearing bones and/or sites with residual disease, postponing radiotherapy to other sites or even omitting it. Due to a divergence in the interpretation of what is feasible, the radiotherapy delivered to this patient group continues to vary between different clinicians and different centers. The recently published outcomes from the randomized BERNIE study, which evaluated the use of bevacizumab in combination with standard chemotherapy, demonstrated that, of 102 metastatic RMS patients, only 31 had radiotherapy to all sites; 49 had radiotherapy to some sites; and 22 received no radiotherapy. It is recognized that selection bias is very likely to have occurred, with metastatic radiotherapy given more often to patients with only one or two metastatic sites; nevertheless, a significant improvement in OS was seen in those patients receiving radiotherapy [33,34,35,36]. In addition, with similar inherent biases in the radiotherapy treatment groups as in the BERNIE study, a single-institute review of 59 patients noted that only 29% of patients had radical radiotherapy, treating all sites of the disease, and again that group had a higher 5-year OS of 76% compared with 11.5% for those treated with partial radiotherapy and 0% for those who received no radiation [34]. Liu et al. published a single-center case series of 13 patients with metastatic RMS or Ewing sarcoma (EwS) who received radiotherapy in a systematic manner (>40 Gy) to all metastatic sites, reporting a 5-year OS of 35% and a 5-year local control rate of the metastases of 92% [37]. Another series of six patients with metastatic RMS also irradiated at all metastatic sites (41.4 Gy–50.4 Gy), showed a 100% local control rate, although out-of-field relapses were seen in 50% of the cases, and the median OS remained poor (31.8 months) [38].

The benefit of whole-lung irradiation for patients with lung metastases only (around 22%) is similarly unclear. The COG published a retrospective analysis of 46 patients treated in the IRS-IV study, in which only 25 patients received whole-lung irradiation, having fewer lung recurrences but no significant differences in OS (47% vs. 31%) [39]. Similarly, in the analysis of 50 patients in the EpSSG RMS 2008 study, in which 26 received whole-lung irradiation, the 3-year PFS was improved from 33% to 56%, but the 3-year OS was unchanged [40]. In a report by the CWS group, there was no effect on the OS, EFS, or local relapse rate in the lungs in the presence or absence of whole-lung radiotherapy [41].

In an effort to improve the understanding on the optimal usage of radiotherapy for metastatic RMS and whether this influences survival, the FaR-RMS trial is randomizing RMS patients with a poor prognosis (≥2 prognostic factors), comparing radiotherapy to all sites that can be feasibly treated with locoregional (primary tumor and involved regional nodes) radiotherapy only [9,26].

### 2.4. Definition of Radiotherapy Target Volumes and Margins in FaR-RMS

#### 2.4.1. GTV

For radiotherapy treatment of the primary tumor volume, the Gross Tumor Volume (GTV) at presentation (GTVp_pre) will be delineated (or reconstructed) for all cases; this refers to the extent of disease at diagnosis (prior chemotherapy volume), taking into account changes in the anatomy and organ displacement resulting from chemotherapy-related tumor shrinkage or surgical resection.

For cases receiving definitive primary radiotherapy (including both arms of RT1c), and for those receiving adjuvant radiotherapy in the dose-escalation arm in RT1b, an additional GTV will be defined based on the extent of the residual primary tumor on imaging obtained post-induction chemotherapy (GTVp_post), taking into account changes in the anatomy and organ displacement resulting from chemotherapy-related tumor shrinkage or surgical resection.

The nodal GTV (GTVn) should be delineated based on the gross extent of nodal involvement at diagnosis, taking into account changes in the anatomy and organ displacement resulting from chemotherapy-related tumor shrinkage or surgical resection. For exceptional cases with pathologically enlarged bulky macroscopic residual nodal disease post-induction chemotherapy, an additional boost should be delivered, with this residual disease delineated as GTVn_post.

#### 2.4.2. CTV

Clinical Target Volumes (CTVs) for the primary tumor (CTVp) will be generated using the following margins:-GTVp_pre to CTVp_pre: 1 cm.-For extremity primary tumor sites, superior and inferior CTV margins of 2 cm are required, with 1 cm expansion circumferentially.-Skin, scar, drain, or biopsy sites should not be included in the CTVp, except in cases of involvement with gross tumor.-GTVp_post to CTVp_post: 0.5 cm.-For tumors arising adjacent to body cavities (e.g., thorax, abdomen, pelvis) that extend or ‘push’ into the cavity but do not infiltrate adjacent organs or tissues, the GTVp should only be expanded, by 1 cm (GTVp_pre) or 0.5 cm (GTVp_post), in the direction of potential infiltration, and there should be no extension of the CTVp into the adjacent, uninvolved body cavity.-GTVn to CTVn: 3 cm superiorly and inferiorly (or in the direction of nodal drainage), and circumferentially to include adjacent lymph nodes in the anatomically constrained lymph node site. Wherever possible, displaced normal tissue should be excluded from the CTVn. In cases of uncertainty, or particular concern, about the exact extent of nodal involvement at diagnosis, an involved field concept should be used.-For bulky residual involved lymph nodes, GTVn_post to CTVn_post: 0.5 cm.

#### 2.4.3. ITV

At primary tumor sites in which respiratory-related motion needs to be considered (e.g., thorax, upper abdomen), the use of 4DCT and an Internal Target Volume (ITV) approach is allowed, based on local practice. This will be denoted as ITVp.

#### 2.4.4. PTV

Expansion from the CTVs or ITVs to the Planning Target Volumes (PTVs) is to be undertaken as per local standard of care, based on the specific radiotherapy technique, image-guidance strategy and set up errors, and is usually in the range of 3 to 10 mm.

### 2.5. Dose Prescription in FaR-RMS

#### 2.5.1. Primary Tumor

According to the FaR-RMS radiotherapy guidelines, a simultaneous integrated boost prescription is allowed. Table 1 shows a summary of the recommendations.

#### 2.5.2. Involved Lymph Nodes

41.4 Gy in 23 fractions over 4.5 weeks (or equivalent) to PTVn.

For bulky residual involved lymph nodes only, Phase 2: 9 Gy in 5 fractions (or equivalent) to PTVn_post.

#### 2.5.3. Metastases

For patients with favorable metastatic disease, with a defined Modified Oberlin Prognostic Score * of ≤1, it is recommended that they continue to receive radiotherapy to all sites of the disease, including all metastases where feasible, as per the current international standard of care.

* Modified Oberlin Prognostic Score (1 point for each adverse factor):-Age ≥ 10 y.-Extremity, other, unidentified primary site.-Bone and/or bone marrow metastatic involvement.-≥3 metastatic organ sites.

Unfavorable metastatic disease: 2–4 adverse factors.

Favorable metastatic disease: 0–1 adverse factors.

Patients with unfavorable metastatic disease, defined as a Modified Oberlin Prognostic Score of ≥2, will all receive radiotherapy to the primary tumor sites (except where unknown) and involved regional lymph nodes; then, they will be randomized to either receive radiotherapy to all sites of metastases, where feasible, or loco-regional radiotherapy only.

A dose of 41.4 Gy in 23 fractions over 4.5 weeks (or equivalent) to metastatic lesions is recommended, with the exception of multiple lung metastases and peritoneal metastases, for which whole-lung irradiation at 15 Gy in 10 fractions and 24 Gy in 16 fractions whole-abdominal irradiation, respectively, are recommended.

### 2.6. Palliative Radiotherapy

In specific cases, such as those with widespread bone and/or bone marrow metastases, palliative radiotherapy can be effectively employed as a symptomatic palliative treatment option, for example, as pain relief for patients with bone metastases or for cases with spinal cord compression. The published data from adult patients support the use of a single fraction of 8 Gy, which provides pain relief equivalent to that of multi-fraction regimes for bone metastases, although a more fractionated approach, such as 20 Gy in 5 fractions or 30 Gy in 10 fractions, is recommended for those with nerve root compression [42,43]. In patients with symptomatic recurrence at a previously irradiated site, retreatment is feasible and can be considered.

## 3. Advanced External Beam Radiotherapy Techniques and Other Modalities

### 3.1. Stereotactic Body Radiotherapy

Stereotactic body radiotherapy (SBRT) precisely targets tumors with very high doses of radiation, often administered in a reduced number of fractions. Also known as stereotactic ablative radiotherapy and stereotactic ablative body radiation (SABR), SBRT can be considered as an option for patients with oligometastatic disease and is permitted in the FaR-RMS study [44].

### 3.2. Particle Therapy

Decisions regarding the use of proton therapy in pediatric RMS vary between different countries depending on their resources, reimbursement criteria, and treatment philosophies. For most countries, this decision is made on a case-by-case basis, investigating the clinically relevant differences between proton and photon therapy in each specific case. Several factors, such as tumor location, patient age, treatment goals, prognosis, and available resources, are taken into account. Proton therapy is a valuable treatment option as it can potentially reduce the risk of long-term side effects in certain patients, particularly those with localized disease [45]. However, for patients with widespread metastatic disease and a poor prognosis, treatment with protons is unlikely to result in any meaningful benefit and can be disadvantageous due to longer treatment delivery time every day; delay in treatment start due to limited capacity; and requiring significantly longer travel and time away from home for some patients and their families.

### 3.3. Brachytherapy

Brachytherapy (BT) is a radiotherapy modality that deploys sealed radioactive sources that are placed at or close to the site to be treated. BT is a well-recognized alternative radiotherapy modality in the treatment of adult sarcoma patients, and its use as a part of the local treatment of RMS has increased over the past decades [46,47,48,49,50,51,52,53,54,55,56,57,58,59]. Its dosimetric advantages can have an impact on the reduction of late toxicity in pediatric RMS survivors, although careful selection of patients and highly specialized experienced teams are needed to make this treatment a success (Figure 2). Detailed guidelines were developed for the FaR-RMS trial and have been recently published [60]. The FaR-RMS trial incorporates the prospective collection of dosimetric brachytherapy data through the SIOP Europe-EORTC QUARTET platform for prospective and retrospective radiotherapy quality assurance [22,23].

### 3.4. Motion Management

Different strategies and techniques are employed to address the challenge of dealing with patient motion during treatment delivery. Managing this motion is particularly important in moving targets located in the chest and abdomen, where the respiratory motion or involuntary movement can significantly affect the accuracy of radiotherapy delivery. A precise control of motion facilitates the use of smaller margins, thus contributing to minimizing the dose to the uninvolved healthy tissue/organs at risk in close proximity to the tumor. Techniques that can be used for motion management, beyond the use of 4DCT to allow the creation of an ITVp encompassing the full extent of respiratory-related motion (previously discussed), are listed below:

**Surface-Guided Radiation Therapy (SGRT)** is a technique that uses stereo vision technology to track the surface of the patients in 3D, for both set up and motion management during RT. Unlike traditional methods that exclusively rely on internal imaging like cone beam CT (CBCT) or oblique X-rays, SGRT uses real-time tracking of the patient’s external surface throughout treatment. It allows real-time motion management, contributing to increased precision, efficiency, patient comfort, and safety.

**Gating** is a technique used to synchronize the delivery of radiation with the patient’s respiratory motion. Gating helps ensure that the radiation beam is only delivered when the tumor is in the optimal position, minimizing the risk of irradiating surrounding healthy tissues.

**Tracking**, on the other hand, involves the continuous monitoring of the position and movement of the tumor or surrounding anatomical structures during treatment delivery, utilizing a number of different strategies such as 1. placement of an X-ray visualized fiducial marker or a radio-frequency transponder in the tumor, 2. the use of ultrasound scan for motion control, or 3. the more accurate 4D cine MRI acquired during MR-guided radiation therapy.

SGRT, gating, and tracking have progressively become relevant tools in modern (pediatric) radiation oncology and are often used in daily practice, although the use of gating can be challenging in very young patients due to the active patient collaboration that is required.

### 3.5. Magnetic Resonance Imaging-Guided Linear Accelerator (MRI-LINAC) Radiotherapy

MRI-LINAC radiotherapy is an advanced medical technology that combines two powerful medical devices: a linear accelerator (LINAC) and a magnetic resonance imaging (MRI) scanner. The radiation delivery on the MRI-LINAC is fully integrated with the MRI. This integration allows for precise and real-time MR image guidance during radiation therapy treatment. Based on its increased precision, it can offer advantages over traditional radiotherapy methods by potentially reducing margins, allowing dose escalation while minimizing the dose to the organs at risk. MRI-LINAC radiotherapy has the potential to realize advantages for the treatment of pediatric patients with cancer with more precise tumor targeting, reduced radiation exposure to healthy tissues, and the ability to adapt treatment plans on a daily basis to a child’s changing anatomy throughout the course of radiation. This technology and daily adaptive radiotherapy are exciting advancements with the potential to further improve outcomes and the quality of life for young cancer patients [61]. However, this comes with the cost of a higher demand of personnel required to allow re-delineation and replanning during every radiotherapy session.

## 4. Conclusions

Radiotherapy, as part of the treatment of RMS, has greatly benefited from recent technical developments, including the increased use of image guidance and the precision in the definition of target volumes. The FaR-RMS clinical trial is looking to determine the optimal timing of radiotherapy and the local treatment sequence, as well as the benefit of dose escalation in high-risk cases, through a series of novel randomized questions. In addition, the implementation of the most up-to-date technological advances in radiation oncology, such as proton therapy and brachytherapy, as well as SBRT and diverse motion management strategies, has the potential to significantly improve outcomes for patients with RMS whilst minimizing treatment-related morbidity.

## Figures and Tables

**Figure 1 cancers-17-01618-f001:**
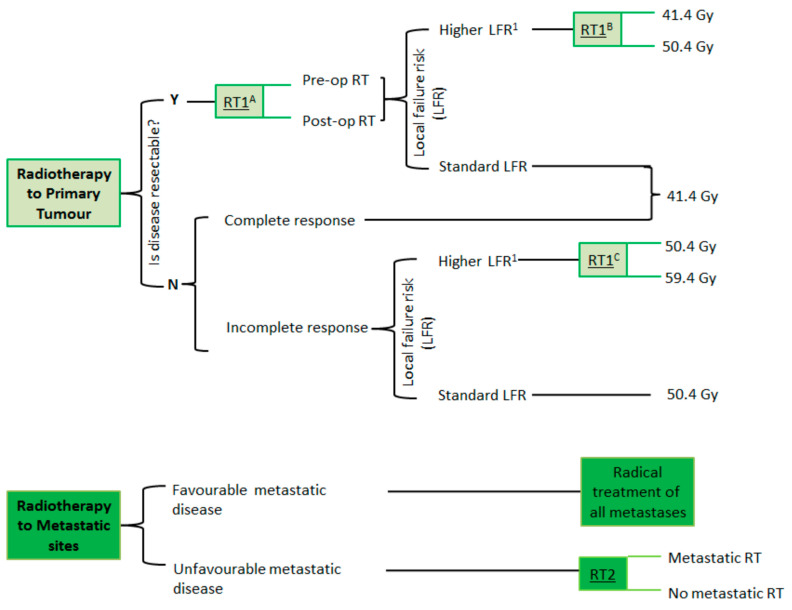
FaR-RMS radiotherapy randomizations.

**Figure 2 cancers-17-01618-f002:**
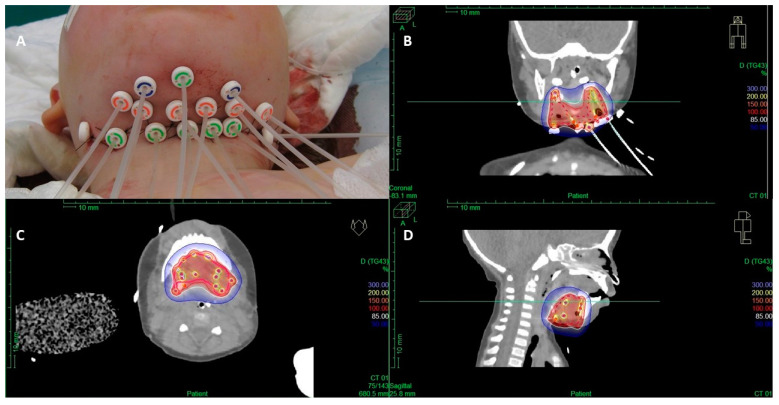
Combined surgical resection and interstitial brachytherapy for the treatment of a floor-of-the-mouth rhabdomyosarcoma (**A**) and dose distribution (**B**–**D**): red isodose surface—100% of the prescribed dose, white—85%, and dark blue—50%; pink dotted line: clinical target volume (CTV).

**Table 1 cancers-17-01618-t001:** Summary of the radiotherapy dose recommendations for primary the tumor within the FaR-RMS trial.

Primary Tumor	Standard Fractionation	Simultaneous Integrated Boost (SIB)
Resectable pre- or post-op radiotherapy HLFR Standard dose	41.4 Gy in 23 fractions over 4.5 weeks to PTVp_pre.	NA
Resectable pre- or post-op radiotherapy HLFR Escalated dose	41.4 Gy in 23 fr over 4.5 weeks to PTVp_Pre 9 Gy in 5 fr to PTVp_Post	42.5 Gy in 28 fr to PTVp_Pre 50.4 Gy in 28 fr to PTVp_post
Resectable pre- or post-op radiotherapy SLFR Standard dose	41.4 Gy in 23 fr over 4.5 weeks to PTVp_pre	NA
Unresectable complete response (to induction chemotherapy) Standard dose	41.4 Gy in 23 fr over 4.5 weeks to PTVp_pre	NA
Unresectable incomplete response (to induction chemotherapy) HLFR Standard dose	41.4 Gy in 23 f over 4.5 weeks to PTVp_Pre 9 Gy in 5 fr to PTVp_Post	42.5 Gy in 28 fr to PTVp_Pre 50.4 Gy in 28 fr to PTVp_post
Unresectable incomplete response (to induction chemotherapy) HLFR Escalated dose	41.4 Gy in 23 fr over 4.5 weeks to PTVp_Pre 18 Gy in 10 fr to PTVp_Post	42.5 Gy in 28 fr to PTVp_Pre 58.1 Gy in 28 fr to PTVp_post
Unresectable incomplete response (to induction chemotherapy) SLFR Standard dose	41.4 Gy in 23 fr over 4.5 weeks to PTVp_Pre 9 Gy in 5 fr to PTVp_Post	42.5 Gy in 28 fr to PTVp_Pre 50.4 Gy in 28 fr to PTVp_post

Abbreviations: fr: fraction. Gy: Gray. HLFR: higher local failure risk. NA: not applicable. SLFR: standard local failure risk. PTVp: planning target volume for primary tumor.

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
