# Peer review of "Current and Future Developments in Radiation Oncology Approach for Rhabdomyosarcoma"

_cancers, 2025, doi:10.3390/cancers17101618_

Round 1
Reviewer 1 Report (Previous Reviewer 2)
Comments and Suggestions for Authors
The author gave a good answer to the questions raised by the reviewers and suggested minor modifications for publication.
1. The first paragraph of the article can add the current situation of cancer treatment and diagnosis and treatment to introduce the RMS that will be discussed in the article. To support this claim, the following published important related papers should be cited: Exploration 2023, 3, 20210111; Coord. Chem. Rev. 2024, 517, 216054.
Author Response
Reviewer 1
Open Review
( ) I would not like to sign my review report
(x) I would like to sign my review report
Quality of English Language
(x) The English could be improved to more clearly express the research.
( ) The English is fine and does not require any improvement.
The English was checked by our last author who is a native speaker of English. We believe further review is not necessary.
Comments and Suggestions for Authors
The author gave a good answer to the questions raised by the reviewers and suggested minor modifications for publication.
- The first paragraph of the article can add the current situation of cancer treatment and diagnosis and treatment to introduce the RMS that will be discussed in the article. To support this claim, the following published important related papers should be cited: Exploration 2023, 3, 20210111; Coord. Chem. Rev. 2024, 517, 216054.
An extensive general introduction on rhabdomyosarcoma specifics, treatment and outcomes was already included (lines 34-78).
Reviewer 2 Report (New Reviewer)
Comments and Suggestions for Authors
This review article outlines recent topics and future prospects in radiation therapy for rhabdomyosarcoma. As a review article, there are certain limitations regarding originality and novelty; however, the content presented is generally accurate. On the other hand, there are insufficient aspects as a review article, and revisions are necessary.
#1. A review article, by definition, summarizes the results of previously published papers. However, this paper contains sections with no citations, such as 3.1 SBRT and 3.2 Particle therapy. While some omissions may be unavoidable, these sections should ideally include citations. A revision from this perspective is requested.
#2. The content appears to be heavily biased toward the FaR-RMS study.
The FaR-RMS study is a pivotal clinical trial; however, it is only one trial, and the description should be more neutral. For details on the FaR-RMS study, please refer to the paper published in Cancers.
Chisholm, J., et al. Frontline and Relapsed Rhabdomyosarcoma (FaR-RMS) Clinical Trial: A Report from the European Paediatric Soft Tissue Sarcoma Study Group (EpSSG). Cancers 2024, 16, 998. DOI: 10.3390/cancers16050998
Author Response
Reviewer 2
Open Review
(x) I would not like to sign my review report
( ) I would like to sign my review report
Quality of English Language
( ) The English could be improved to more clearly express the research.
(x) The English is fine and does not require any improvement.
Comments and Suggestions for Authors
This review article outlines recent topics and future prospects in radiation therapy for rhabdomyosarcoma. As a review article, there are certain limitations regarding originality and novelty; however, the content presented is generally accurate. On the other hand, there are insufficient aspects as a review article, and revisions are necessary.
#1. A review article, by definition, summarizes the results of previously published papers. However, this paper contains sections with no citations, such as 3.1 SBRT and 3.2 Particle therapy. While some omissions may be unavoidable, these sections should ideally include citations. A revision from this perspective is requested.
We thank you the reviewer for pointing this aspect out. To comply with the reviewer’s request, the following references have been included in sections 3.1. Stereotactic body radiotherapy, and 3.2. Particle therapy (lines 282 and 291, respectively):
- Christopher L. Tinkle, Charu Singh, Shane Lloyd, Yian Guo, Yimei Li, Alberto S. Pappo, Steven G. DuBois, John T. Lucas Jr., Daphne A. Haas-Kogan, Stephanie A. Terezakis, Steve E. Braunstein, Matthew J. Krasin. Stereotactic body radiotherapy for metastatic and recurrent solid tumors in children and young adults. Int J Radiat Oncol Biol Phys. 2021 April 01; 109(5): 1396–1405. doi:10.1016/j.ijrobp.2020.11.054.
- Meng Dong, Jianrong Wu, Renhua Wu, Dandan Wang, Ruifeng Liu, Hongtao Luo, Yuhang Wang, Junru Chen, Yuhong Ou, Qiuning Zhang and Xiaohu Wang. Efficacy and safety of proton beam therapy for rhabdomyosarcoma: a systematic review and meta-analysis. Radiat Oncol 18, 31 (2023). https://doi.org/10.1186/s13014-023-02223-6.
#2. The content appears to be heavily biased toward the FaR-RMS study.
The FaR-RMS study is a pivotal clinical trial; however, it is only one trial, and the description should be more neutral. For details on the FaR-RMS study, please refer to the paper published in Cancers.
Chisholm, J., et al. Frontline and Relapsed Rhabdomyosarcoma (FaR-RMS) Clinical Trial: A Report from the European Paediatric Soft Tissue Sarcoma Study Group (EpSSG). Cancers 2024, 16, 998. DOI: 10.3390/cancers16050998
This paper was the result of an invitation to collaborate with Cancers on the Special Issue "Improving Treatment for Rhabdomyosarcoma: A Collaborative Approach through the European Paediatric Soft Tissue Sarcoma Study Group (EpSSG)". Guest Editor(s): Julia Chisholm, Janet Shipley, Johannes H. M Merks.
We were asked to contribute with an overview of the radiotherapy aspects within the EpSSG. Therefore, the content of this manuscript reveals the core aspects of the radiotherapy approach in EpSSG, and specifically within the currently ongoing FaR-RMS trial. Our manuscript is a complement of the appointed by reviewer 2 by Chisholm (DOI: 10.3390/cancers16050998), and the content overlap has been reduced to the bare minimum.
This manuscript is a resubmission of an earlier submission. The following is a list of the peer review reports and author responses from that submission.
Round 1
Reviewer 1 Report
Comments and Suggestions for Authors
Dear authors,
This manuscript represents a true advance in the cancer science, it is high impact the power of chemotherapy, considering all the advantages and disadvantages of this approach on the treatment of a so specific and important cancer tumor.
I only suggest a final review in the English, because I found (not many) some typos,
Thanks loads,
Regards,
Author Response
We would like to thank the reviewers for mentioning interesting issues and posing some questions that helped us to improve the quality of our manuscript.
Changes in the revised manuscript are done according to the reviewers’ suggestions. Please note page and line numbers refer to the manuscript marked version; text additions are indicated in red and removed text is indicated in strikethrough red.
Reviewer 1
Dear authors,
This manuscript represents a true advance in the cancer science, it is high impact the power of chemotherapy, considering all the advantages and disadvantages of this approach on the treatment of a so specific and important cancer tumor.
I only suggest a final review in the English, because I found (not many) some typos,
Thanks loads,
Regards,
Thank you very much. As suggested, a thorough review of the language has been done by the native English speaker author.

Reviewer 2 Report
Comments and Suggestions for Authors
This manuscript introduced radiotherapy and advanced treatment techniques for rhabdomyosarcoma (RMS). RMS is more common in children and adolescents, with a 30% chance of recurrence or metastasis after treatment. Therefore, it is very important to summarize the existing treatment methods and techniques and explore better treatment options. The summary of this manuscript shows the implementation of the most up to date technological advances in radiation oncology, such as proton therapy and brachytherapy has the potential to significantly improve outcomes for patients with RMS whilst minimizing treatment-related morbidity. However, there are still some problems in the manuscript. It is recommended that the manuscript be published in cancers after completing revision.
1. In the simple summary part of this article, RMS appears for the first time, please complete the full name of RMS in this part.
2. The text in figure 1 is blurred, please replace it with a clearer picture.
3. Please change the letters in figure 2 to lower case. And the layout of the picture is not beautiful, please adjust to the appropriate size and position.
4. There are some formatting issues in the article, the need for Spaces between numbers and units, such as “40Gy”.
5. The title of the article is “Current and Future Developments in Radiation Oncology Approach for RMS”, but the article mostly describes the existing treatment options and advanced technologies, and does not mention the future development. Please elaborate.
6. The format of references is not uniform. For example, Ref. 17 should be changed to “Pediatr Blood Cancer, 2014. 61:S133-S134.”; Ref. 19 should be changed to “Radiotherapy and Oncology, 2023. 182:109549.”; Ref. 21 should not refer directly to the URL. Please check the similar issues and revise.
7. The first paragraph of the article can add the current situation of cancer treatment and diagnosis and treatment to introduce the RMS that will be discussed in the article. Here are many recent articles for reference: Exploration 2023, 3, 20210111; Coord. Chem. Rev. 2024, 517, 216054.
Comments on the Quality of English Language
The English could be improved to more clearly express the research.
Author Response
We would like to thank the reviewers for mentioning interesting issues and posing some questions that helped us to improve the quality of our manuscript.
Changes in the revised manuscript are done according to the reviewers’ suggestions. Please note page and line numbers refer to the manuscript marked version; text additions are indicated in red and removed text is indicated in strikethrough red.
Reviewer 2
This manuscript introduced radiotherapy and advanced treatment techniques for rhabdomyosarcoma (RMS). RMS is more common in children and adolescents, with a 30% chance of recurrence or metastasis after treatment. Therefore, it is very important to summarize the existing treatment methods and techniques and explore better treatment options. The summary of this manuscript shows the implementation of the most up to date technological advances in radiation oncology, such as proton therapy and brachytherapy has the potential to significantly improve outcomes for patients with RMS whilst minimizing treatment-related morbidity. However, there are still some problems in the manuscript. It is recommended that the manuscript be published in cancers after completing revision.
- In the simple summary part of this article, RMS appears for the first time, please complete the full name of RMS in this part.
The sentences has been adjusted as suggested (lines 21-23): “However, about one third of the rhabdomyosarcoma (RMS) patients will develop loco-regional relapse, and patients with metastatic or relapsed disease have poor prognosis, so improved local therapies are needed, in addition to better systemic therapies.”
- The text in figure 1 is blurred, please replace it with a clearer picture.
As suggested the resolution of the image has been increased. (Lines 107-8).
- Please change the letters in figure 2 to lower case. And the layout of the picture is not beautiful, please adjust to the appropriate size and position.
We are sorry that the format of reviewer 2 is off. In the version of manuscript we are currently working the case is the appropriate and the image is centered.
- There are some formatting issues in the article, the need for Spaces between numbers and units, such as “40Gy”.
Thank you for suggesting this. The format has been changed accordingly.
- The title of the article is “Current and Future Developments in Radiation Oncology Approach for RMS”, but the article mostly describes the existing treatment options and advanced technologies, and does not mention the future development. Please elaborate.
The future developments are describes in section 3. A number of the techniques described there are not current practice in pediatric radiotherapy yet.
- The format of references is not uniform. For example, Ref. 17 should be changed to “Pediatr Blood Cancer, 2014. 61:S133-S134.”;Ref. 19 should be changed to “Radiotherapy and Oncology, 2023. 182:109549.”; Ref. 21 should not refer directly to the URL. Please check the similar issues and revise.
The format been changed accordingly.
- The first paragraph of the article can add the current situation of cancer treatment and diagnosis and treatment to introduce the RMS that will be discussed in the article. Here are many recent articles for reference: Exploration 2023, 3, 20210111; Coord. Chem. Rev. 2024, 517, 216054.
The first paragraph has been rephrased as follows: (lines 34-44)
‘Rhabdomyosarcoma (RMS) is an infrequent soft tissue sarcoma that mostly presents in children (around 60% of cases), while the adult cases carry a poorer prognosis [1, 2]. RMS is the commonest of the pediatric soft tissue sarcomas, and affects around 40 children (0-14 years) and 50 teenagers/adults per year in the UK [1,3]. RMS derives from embryonal mesenchyme and can arise in almost all different sites within the human body. The most frequent histological sub-types are: alveolar (aRMS), embryonal (eRMS) and spindle cell/ sclerosing (ssRMS) [4]. Neoadjuvant chemotherapy is used in the majority of patients leading to a response rate (RR) of around 80-85% [5-7].’

Reviewer 3 Report
Comments and Suggestions for Authors
This manuscript provides a thorough overview of radiation oncology approaches in paediatric Rhabdomyosarcoma (RMS), including current practices, multimodality treatment strategies, and approaches for metastatic disease. It highlights the chemosensitive nature of RMS and integrates discussions on radiotherapy modalities such as stereotactic body therapy, particle therapy, and proton therapy, with concise descriptions of their usage and limitations. The manuscript also mentions key trials, including the EpSSG and COG studies, with a notable summary of the FaR-RMS trial, which adds significant value.
Strengths
1. Clarity and Scope: The manuscript is well-structured, clearly defines its scope, and focuses on paediatric RMS while noting the dearth of literature on adult RMS.
2. Comprehensive Coverage: It thoroughly discusses the current state of radiotherapy, including multimodality treatment approaches (neoadjuvant and adjuvant therapies).
3. Up-to-Date Literature: The literature cited is current and includes major trials such as EpSSG and COG studies, making the discussion relevant and evidence-based.
4. Metastatic RMS: The approach to metastatic RMS is well-covered, providing valuable insights into this challenging subset. The manuscript is well-written, informative, and evidence-based. It will be a valuable addition to literature in relation to the current understanding of RMS management.
Please include the following minor comments:
Abstract: Please provide a structured abstract. It is too brief.
Introduction: What did you hypothesize before conducting this study? Please provide your hypothesis in 2-3 lines at the end of the Introduction section.
Discussion: Please add limitations of this manuscript.
Author Response
We would like to thank the reviewers for mentioning interesting issues and posing some questions that helped us to improve the quality of our manuscript.
Changes in the revised manuscript are done according to the reviewers’ suggestions. Please note page and line numbers refer to the manuscript marked version; text additions are indicated in red and removed text is indicated in strikethrough red.
Reviewer 3
This manuscript provides a thorough overview of radiation oncology approaches in paediatric Rhabdomyosarcoma (RMS), including current practices, multimodality treatment strategies, and approaches for metastatic disease. It highlights the chemosensitive nature of RMS and integrates discussions on radiotherapy modalities such as stereotactic body therapy, particle therapy, and proton therapy, with concise descriptions of their usage and limitations. The manuscript also mentions key trials, including the EpSSG and COG studies, with a notable summary of the FaR-RMS trial, which adds significant value.
Strengths
- Clarity and Scope: The manuscript is well-structured, clearly defines its scope, and focuses on paediatric RMS while noting the dearth of literature on adult RMS.
- Comprehensive Coverage: It thoroughly discusses the current state of radiotherapy, including multimodality treatment approaches (neoadjuvant and adjuvant therapies).
- Up-to-Date Literature: The literature cited is current and includes major trials such as EpSSG and COG studies, making the discussion relevant and evidence-based.
- Metastatic RMS: The approach to metastatic RMS is well-covered, providing valuable insights into this challenging subset. The manuscript is well-written, informative, and evidence-based. It will be a valuable addition to literature in relation to the current understanding of RMS management.
Please include the following minor comments:
Abstract: Please provide a structured abstract. It is too brief.
The abstract has been written in line with the perspective papers of the other papers included in the RMS issue (see doi: 10.3390/cancers15020449 and https://doi.org/10.3390/cancers16050998)
Introduction: What did you hypothesize before conducting this study? Please provide your hypothesis in 2-3 lines at the end of the Introduction section.
The aim of the paper is described in sections Brief summary and abstract (lines 24-26 and 28-31
“This review aims to highlight the current and future developments in radiotherapy as part of the management of RMS, which have the objective of improving outcomes whilst minimizing treatment-related morbidity.”
“The current thinking and future developments in the radiation oncology field about how to raise cure rates, especially in the highest-risk patients are presented.”
As to the aim of the FaR-RMS study hypothesis, this has been extensively described in https://doi.org/10.3390/cancers16050998 (included as reference in order to avoid repetition).

Reviewer 4 Report
Comments and Suggestions for Authors
In this manuscript the authors review the current and future developments in radiotherapy treatments of rhabdomyosarcoma.
However, there are aspects that need to be clarified:
1) from the Introduction it is not clear what is the novelty of this review with respect to other review studies in the literature (some of which cover also radiotherapy approaches for RMS); for example: https://pmc.ncbi.nlm.nih.gov/articles/PMC10650215/
https://www.frontiersin.org/journals/oncology/articles/10.3389/fonc.2019.01458/full
https://www.cclg.org.uk/write/MediaUploads/Events/Sarcoma%20Course%20May%202022/Henry_Mandeville_-_Radiotherapy_in_the_Management_of_Childhood_Rhabdomyosarcoma_Clinical_Oncology_2019.pdf
https://www.frontiersin.org/journals/oncology/articles/10.3389/fonc.2022.1016894/full
https://www.nature.com/articles/s41572-018-0051-2
and many more...
So what is the new approach/knowledge/etc. of this review compared to other reviews in the literature?
2) The Conclusion is very "inconclusive" for this reviewer. There are only 3 sentences: a first very generic sentence about the importance of radiotherapy for RMS treatment ; a 2nd sentence about the FaR-RMS clinical trial ; and a 3rd generic sentence about the potential of new treatment approaches (proton therapy and brachytherapy). What about the therapies mentioned in sections 3.1, 3.4, 3.5?
Overall, this review presents a summary of information, with no in-depth discussion of this information, of the future perspectives in the field... Where will the field move in the next 10 years or so? (given the huge advancements in any areas driven, for example, by the advancements in computer power, AI, all with the purpose of advancing personalised medicine, ...) There is no "personal" discussion at the end of this review highlighting authors' view of the field, their own perspectives, etc...
Author Response
We would like to thank the reviewers for mentioning interesting issues and posing some questions that helped us to improve the quality of our manuscript.
Changes in the revised manuscript are done according to the reviewers’ suggestions. Please note page and line numbers refer to the manuscript marked version; text additions are indicated in red and removed text is indicated in strikethrough red.
Reviewer 4
In this manuscript the authors review the current and future developments in radiotherapy treatments of rhabdomyosarcoma.
However, there are aspects that need to be clarified:
1) from the Introduction it is not clear what is the novelty of this review with respect to other review studies in the literature (some of which cover also radiotherapy approaches for RMS); for example: https://pmc.ncbi.nlm.nih.gov/articles/PMC10650215/
https://www.frontiersin.org/journals/oncology/articles/10.3389/fonc.2019.01458/full
https://www.cclg.org.uk/write/MediaUploads/Events/Sarcoma%20Course%20May%202022/Henry_Mandeville_-_Radiotherapy_in_the_Management_of_Childhood_Rhabdomyosarcoma_Clinical_Oncology_2019.pdf
https://www.frontiersin.org/journals/oncology/articles/10.3389/fonc.2022.1016894/full
https://www.nature.com/articles/s41572-018-0051-2
and many more...
So what is the new approach/knowledge/etc. of this review compared to other reviews in the literature?
What makes this article to stand out is the focus on the radiotherapy aspects related to the treatment of rhabdomyosarcoma. It presents an overview about the current ongoing trial of the European paediatric Soft Tissue Sarcoma Study Group, the FaR-RMS study, an overarching study for children and adults with Frontline and Relapsed RhabdoMyoSarcoma trial. The specificities of this trial, including radiotherapy randomizations have not been published so far.
2) The Conclusion is very "inconclusive" for this reviewer. There are only 3 sentences: a first very generic sentence about the importance of radiotherapy for RMS treatment ; a 2nd sentence about the FaR-RMS clinical trial ; and a 3rd generic sentence about the potential of new treatment approaches (proton therapy and brachytherapy). What about the therapies mentioned in sections 3.1, 3.4, 3.5?
The conclusion has been rephrased as follows: (lines 354-8)
“In addition, the implementation of the most up to date technological advances in radiation oncology, such as proton therapy and brachytherapy, as well as SBRT and diverse motion management strategies, have the potential to significantly improve outcomes for patients with RMS whilst minimizing treatment-related morbidity.”
Overall, this review presents a summary of information, with no in-depth discussion of this information, of the future perspectives in the field... Where will the field move in the next 10 years or so? (given the huge advancements in any areas driven, for example, by the advancements in computer power, AI, all with the purpose of advancing personalised medicine, ...) There is no "personal" discussion at the end of this review highlighting authors' view of the field, their own perspectives, etc...
We really appreciate the feedback and big effort of the reviewers during the review process to help us improve the manuscript.

Round 2
Reviewer 4 Report
Comments and Suggestions for Authors
I have downloaded the new version of the manuscript, but I do not see the responses to reviewers comments incorporated into the manuscript.
1) The originality/novelty of this study, compared to the published literature (that contains studies reviewing radiotherapy treatments for RMS) needs to be stated in very clear sentences in the main text (not in the online version of the Response to Reviewers).
2) Also, I do not see any significant changes in the Conclusion section (apart from a half a sentence in red in which the authors added 2 more therapies that they mentioned in the above sections).
This was not my original comment: I wanted to see an in-depth discussion and conclusion of this study. How does it contribute to the current knowledge, to the already published literature? So that the originality/novelty of this study is emphasised again!
Round 3
Reviewer 4 Report
Comments and Suggestions for Authors
---